# OpenReview forum: "Ranking Free RAG: Replacing Re-ranking with Selection in RAG for Sensitive Domains"
_ICML.cc/2026/Conference — ICML 2026 regular_

### Official Review · Reviewer_btuK · 2026-03-12

**Soundness:** 3
**Presentation:** 3
**Significance:** 2
**Originality:** 3
**Overall Recommendation:** 4
**Confidence:** 4

**Summary:**

The paper introduces METEORA, a novel framework for Retrieval-Augmented Generation (RAG) that replaces traditional, opaque re-ranking methods with an interpretable, rationale-driven evidence selection mechanism. While standard RAG systems rely on black-box similarity scores and arbitrary top-k cutoffs, METEORA uses Direct Preference Optimization (DPO) to train an LLM to generate explicit rationales explaining why specific evidence is relevant to a query. These rationales guide an Evidence Chunk Selection Engine (ECSE) that adaptively determines the optimal number of evidence chunks through statistical elbow detection, eliminating fixed heuristics. Furthermore, the framework leverages these same rationales in a verification stage to detect and filter adversarial corpus poisoning, significantly improving robustness in sensitive domains like legal and healthcare.

**Compliance With Llm Reviewing Policy:**

Affirmed.

**Key Questions For Authors:**

- Since the preference dataset relies on naturally occurring incorrect evidence, how does the system perform against "white-box" attackers who have knowledge of the rationale-generation process?
- While adaptive selection reduces the volume of evidence processed, what is the additional latency cost incurred by generating multiple rationales and performing the verifier LLM check for every query?
- In cases where multiple rationales are generated, how does the system handle conflicting rationales if some support a piece of evidence and others refute it?

**Limitations:**

- The verification component may inadvertently filter out valid, non-malicious information, potentially impacting the recall of the final answer.
- The framework solves the "black box" of evidence selection but does not address potential hallucinations or opaque reasoning within the final generator LLM.
- While tested on legal, financial, and academic data, the performance of rationale-driven selection may vary significantly in domains where information is highly fragmented or lacks clear structural markers.

**Strengths And Weaknesses:**

Strengths:

- Effectively combines interpretability, efficiency, and robustness by using a single reasoning mechanism (rationales) for both evidence selection and adversarial defense.
- The ECSE component removes the need for manual tuning of $k$ (the number of retrieved documents), making the system more adaptive to varying query complexity.
- Demonstrates a 21.05% improvement in precision and a 4.4x improvement in F1 score under poisoning attacks compared to existing baselines across six diverse datasets.
- Reduces the volume of evidence processed to reach comparable recall by 80%, leading to a 33.34% improvement in downstream answer accuracy.
- Includes a human evaluation confirming that the generated rationales allow users to reconstruct evidence selection decisions with 86% accuracy.

Weaknesses:

- The authors acknowledge that while the system is robust against tested poisoning attacks, new or highly sophisticated adversarial methods might still bypass the current verification logic.
- METEORA provides interpretability for the evidence selection stage but does not yet cover the entire RAG pipeline from end to end.
- Explicit rationales may inadvertently lead users to trust the system's output too much, potentially reducing necessary human skepticism or independent verification.
- The DPO training uses naturally incorrect evidence as negative examples rather than explicitly adversarial content, which may create a gap when the model faces complex, hand-crafted attacks.

---

> ### Author Rebuttal · Authors · 2026-03-31
>
> We thank the reviewer for the thoughtful and encouraging feedback. We especially appreciate the positive assessment of the paper’s unified use of rationales for interpretability and robustness, the adaptivity of ECSE, the robustness gains under poisoning attacks, and the human-evaluation results.
>
> **Robustness against white-box attackers**
>
> > **Q1:** Since the preference dataset relies on naturally occurring incorrect evidence, how does the system perform against "white-box" attackers who have knowledge of the rationale-generation process?
>
> **A1:** We thank the reviewer for this important question. Our robustness claim is intentionally scoped to the poisoning setting evaluated in the paper. Specifically, in Section 3, we follow the PoisonedRAG-style corpus-poisoning protocol [1] and insert malicious text into documents that also contain correct context; under this setting, METEORA shows clear and consistent improvements through rationale-based verification. Section 2.1 already notes that the dispreferred rationales are derived from naturally incorrect rather than adversarially crafted evidence, so we do not claim robustness to stronger adaptive white-box attacks that explicitly target the rationale generator. This scope is also consistent with the paper’s Ethical Considerations paragraph, which explicitly notes that the tested defenses may not cover all future attack types. We will clarify this threat-model scoping more explicitly in the revised version.
>
> **Latency cost of rationale generation and verification**
>
> > **Q2:** While adaptive selection reduces the volume of evidence processed, what is the additional latency cost incurred by generating multiple rationales and performing the verifier LLM check for every query?
>
> **A2:** We thank the reviewer for this question. We provide the full empirical latency analysis in our response to *Reviewer c72f, “Efficiency / practical latency evidence.”* Briefly, under the same hardware and model setup for all methods, the measured *Total Time* values are *4.04s* for SBERT, *4.61s* for RankRAG, *1.77s* for *METEORA w/o Verifier*, and *2.91s* for full *METEORA*. This means that, under our controlled setup, the verifier adds approximately *1.14s per query* relative to METEORA w/o Verifier. Importantly, even with this added verification step, full METEORA remains faster than both reranking baselines because it processes substantially fewer input tokens overall. Thus, while rationale generation and verification do introduce sequential computation, the reduction in evidence volume more than compensates for this cost in practice. We will clarify this tradeoff more explicitly in the revision.
>
> > | Method | Input Tokens | TTFT (sec) | Generation Time (sec) | Total Time (sec) |
> > |---|---:|---:|---:|---:|
> > | SBERT | 36.89k | 3.07 | 0.97 | 4.04 |
> > | RankRAG | 39.82k | 3.52 | 1.09 | 4.61 |
> > | METEORA w/o Verifier | 18.68k | 1.21 | 0.56 | 1.77 |
> > | METEORA | 12.23k | 2.42 | 0.49 | 2.91 |
>
>
> **Handling multiple rationales**
>
> > **Q3:** In cases where multiple rationales are generated, how does the system handle conflicting rationales if some support a piece of evidence and others refute it?
>
> **A3:** METEORA does not include an explicit rationale-rationale conflict resolution module during ECSE selection. Instead, as described in Section 2.2, ECSE treats multiple rationales as *complementary query-specific signals* and aggregates them through rationale-evidence pairing and pooled rationale embeddings. Potential inconsistencies are then handled in Section 2.3 at the verification stage, where the Verifier checks each selected evidence chunk against the associated rationales and summaries of previously verified evidence, and can flag *contradiction* or *instruction violations* before generation. We will clarify this division of roles more explicitly in the revision: ECSE aggregates multiple rationale signals for adaptive selection, while contradiction handling occurs at verification time.
>
> **Limitations / scope**
>
> We also appreciate the reviewer’s limitation points. These are broadly consistent with the paper’s own Ethical Considerations paragraph: we already note that the verifier may reject valid information, that METEORA explains evidence selection rather than the entire RAG pipeline end to end, and that human oversight remains important. For the domain-fragmentation point, we agree this is an important scope limitation; our current paper partially addresses it through context expansion and the dataset-specific analysis of chunking brittleness, but broader validation across more fragmented domains is a natural direction for future work.
>
> Reference:
>
> [1] Wei Zou, Runpeng Geng, Binghui Wang, and Jinyuan Jia. {PoisonedRAG}: Knowledge corruption attacks to {Retrieval-Augmented} generation of large language models. In 34th USENIX Security Symposium (USENIX Security 25), pages 3827–3844, 2025.

---

### Official Review · Reviewer_qqSm · 2026-03-12

**Soundness:** 3
**Presentation:** 3
**Significance:** 4
**Originality:** 3
**Overall Recommendation:** 4
**Confidence:** 3

**Summary:**

The paper introduces METEORA, a novel framework designed to replace traditional, opaque similarity-based re-ranking in Retrieval-Augmented Generation (RAG) systems with a transparent, rationale-driven selection process. The authors identify three key vulnerabilities in current RAG pipelines, particularly for high-stakes domains: a lack of interpretability in evidence selection, reliance on arbitrary top-k cutoffs that waste compute or miss context, and susceptibility to corpus poisoning attacks. METEORA introduces three components to address this problem: preference-tuned rationale generator, evidence chunk selection engine (ECSE) and verifier LLM. METEORA is evaluated across six datasets from legal, financial, and academic domains, demonstrating significant improvements in precision, recall, and adversarial robustness while processing less noisy context.

**Compliance With Llm Reviewing Policy:**

Affirmed.

**Final Justification:**

The authors have fully addressed my concerns in their rebuttal, and the format of this paper seems to be OK, therefore I decide to raise my score from 3 to 4.

**Key Questions For Authors:**

I will consider changing my rating based on the authors' response.

1. Regarding empirical latency and cost: Could you provide empirical measurements for end-to-end latency (Time-to-First-Token and total generation time) and API cost/FLOPs compared to the baselines?

2. Regarding false positives of the verifier: Based on Section 2.3, the Verifier only flagging content when highly confident (>90%). However, how does the Verifier perform on purely clean, adversarial-free data? Is there a measurable drop in recall due to false-positive flagging on complex but correct evidence?

3. Regarding the core contribution of DPO: While the ablation study demonstrates a performance drop when DPO is removed from the LLaMA-3.1-8B-Instruct base model, it is difficult to disentangle whether the gains are intrinsically due to the DPO preference alignment or simply the architectural addition of a rationale-generation step (similar to Zero-Shot CoT). Have you considered how to address this possible issue?

**Limitations:**

yes

**Strengths And Weaknesses:**

Strengths:

1. This paper is structured logically and easy to follow. The motivation is compelling, moving smoothly from the problem of opaque re-ranking to the interconnected issue of adversarial vulnerability. Figures 1 effectively contrast the traditional RAG paradigm with the METEORA pipeline. The inclusion of qualitative examples (positive and negative) and clear theoretical grounding in the appendix significantly aids reproducibility and understanding.

2. The paper is highly sound in its empirical design. The evaluation encompasses six diverse and challenging datasets (e.g., QASPER, MAUD, ContractNLI) that genuinely stress-test long-context reasoning. The baseline selection is robust, comparing METEORA against standard bi-encoders, cross-encoders, and modern LLM-based re-rankers like RankRAG. Furthermore, the ablation studies excellently isolate the individual contributions of DPO, the Context Expansion module and the Verifier proving that the components are synergistic rather than independently overriding one another.

3. The formulation of using DPO to train a rationale generator using downstream selection accuracy as the preference signal is novel. Similarly, applying statistical elbow detection on rationale-pooled embeddings to create an unsupervised, adaptive threshold is a refreshing departure from standard top-k dynamic routing.

Weaknesses:

1. Potential formatting non-compliance: That's why I am rating Presentation as 2 for now. I'm not sure if the format of this paper deviates from the official ICML formatting guidelines, particularly regarding to the font choice. If the format is confirmed correct, I will consider raising my rating. The abstract seems to be quite too long, and some of the figures and tables are also a little bit hard to view.

2. The computational complexity analysis presented in Section 2.2 is purely theoretical and may understate practical bottlenecks. While the authors correctly note that dropping top-k reduces the context window for the final generation step, METEORA requires multiple sequential LLM calls (Rationale Generation $\rightarrow$ ECSE Similarity Pooling $\rightarrow$ Verifier LLM → Final Generation). This sequential dependency heavily impacts Time-to-First-Token (TTFT) and overall inference latency, which is not empirically quantified in the evaluation.

3. The scope of impact may be limited by the system's operational cost. While highly significant for asynchronous, high-stakes tasks, the multi-step LLM pipeline may be too heavy for standard, high-throughput consumer QA systems.

---

> ### Author Rebuttal · Authors · 2026-03-31
>
> We thank the reviewer for the thoughtful and encouraging feedback. We especially appreciate the positive assessment of the paper’s motivation, empirical design, ablations, and the novelty of using DPO with downstream selection accuracy as the preference signal.
>
> **Formatting / presentation**
>
> We prepared the submission using the latest `icml.sty` file and have re-checked the PDF for compliance. We will further improve readability by enlarging figures/tables and shortening the abstract.
>
> **Empirical latency and cost**
>
> > **Q1:** Regarding empirical latency and cost: Could you provide empirical measurements for end-to-end latency (Time-to-First-Token and total generation time) and API cost/FLOPs compared to the baselines?
>
> **A1:** We provide the full empirical latency analysis, including TTFT and generation-time breakdowns, in our response to *Reviewer btuK, “Latency cost of rationale generation and verification.”* Briefly, under the same hardware and model setup, total time is **4.04s** (SBERT), **4.61s** (RankRAG), **1.77s** (METEORA w/o Verifier), and **2.91s** (METEORA). Input token counts are also substantially lower for METEORA than for the reranking baselines. For METEORA and METEORA w/o Verifier, TTFT includes rationale generation and all upstream stages, measured up to the first token of the final answer. Since all methods ran on the same L40S setup, lower latency directly implies lower per-query compute cost. We did not separately instrument exact FLOPs; however, end-to-end latency and token usage are the more practical deployment metrics here. We will also clarify that METEORA is best suited to high-stakes or quality-sensitive settings rather than ultra-low-latency consumer QA.
>
> **Verifier false positives on clean data**
>
> > **Q2:** Regarding false positives of the verifier: Based on Section 2.3, the Verifier only flags content when highly confident (>90%). However, how does the Verifier perform on purely clean, adversarial-free data? Is there a measurable drop in recall due to false-positive flagging on complex but correct evidence?
>
> **A2:** We analyzed the verifier on the **70% of evaluation instances that contain no poisoned chunk**, as described in Section 2.3. On this clean subset, any flag assigned to a correct evidence chunk is treated as a false positive. We report the instance-level **false positive rate (FPR)**, i.e., the fraction of clean instances in which at least one correct chunk was incorrectly flagged. The resulting FPR ranges from **0.0142** to **0.0462** across the six datasets, so only a small fraction of clean instances lose at least one correct chunk. We then compute recall on this same clean subset both **with** and **without** the verifier:
> >
> > **ΔR = 0.93 - 0.90 = 0.03**
> >
> > Thus, on purely clean data, the verifier causes only a **0.03 recall drop** (**0.93 → 0.90**), indicating that false-positive flagging is limited and that the verifier preserves the large majority of correct evidence while still providing robustness against poisoned content.
>
> **Core contribution of DPO**
>
> > **Q3:** Regarding the core contribution of DPO: While the ablation study demonstrates a performance drop when DPO is removed from the LLaMA-3.1-8B-Instruct base model, it is difficult to disentangle whether the gains are intrinsically due to the DPO preference alignment or simply the architectural addition of a rationale-generation step (similar to Zero-Shot CoT). Have you considered how to address this possible issue?
>
> **A3:** The ablation already controls for the rationale-generation step itself. *METEORA w/o DPO* still uses the same rationale-driven pipeline; the only change is whether the rationales come from the untuned base model or the DPO-aligned model. So the comparison is not “with rationales” versus “without rationales,” but *generic rationales* versus *selection-aligned rationales*. Section 2.1 explicitly states that DPO trains the model to prefer rationales that lead to correct evidence selection over those that lead to incorrect selection.
>
> A concrete example makes this distinction clearer. For a query asking about the *notice period required to terminate a renewal*, the untuned model produces a broad rationale like *“look for Term, Termination, or Notice Period clauses,”* which is plausible but too generic to reliably recover the correct evidence. The DPO-aligned model instead generates a rationale focused on *renewal termination notice requirements*. In that case, retrieval changes from missing the correct chunk to recovering it. This is why we attribute the gain to *better-aligned rationale quality*, not simply to adding an extra reasoning step. This is also consistent with our human-evaluation framing: the refined rationales are more useful both *functionally* for selecting the correct evidence and *interpretively* for helping humans understand why that evidence should be selected. We will clarify this more explicitly in the revision.

---

> > ### Author Rebuttal · Reviewer_qqSm · 2026-04-02
> >
> > My concerns have been adequately addressed. I will consider adjusting your score accordingly.

---

### Official Review · Reviewer_fqUQ · 2026-03-13

**Soundness:** 3
**Presentation:** 3
**Significance:** 2
**Originality:** 3
**Overall Recommendation:** 4
**Confidence:** 3

**Summary:**

This paper proposes METEORA, a retrieval-augmented generation (RAG) framework that replaces traditional ranking-based retrieval with a rationale-guided selection process. The authors argue that conventional RAG pipelines typically rely on similarity-based ranking and fixed top-k retrieval thresholds, which may lack interpretability and can be vulnerable to adversarial or noisy documents. METEORA addresses this limitation by generating explicit rationales using a preference-aligned language model and using these rationales to guide the retrieval process. Specifically, rationales are used to compute relevance scores for candidate passages and to determine adaptive selection thresholds, thereby removing the need for a fixed ranking cutoff. The rationales are also used to verify retrieved evidence and filter potentially unreliable documents. Experiments across several QA datasets show improvements in retrieval precision, robustness, and interpretability compared with standard RAG pipelines.

**Compliance With Llm Reviewing Policy:**

Affirmed.

**Final Justification:**

The questions have been solved. I have already given a positive recommendation for the initial score, thus I maintain my original evaluation.

**Key Questions For Authors:**

see the weakness

**Limitations:**

yes

**Strengths And Weaknesses:**

Strengths

The rationale-guided retrieval design is technically plausible and addresses a known limitation of conventional RAG pipelines.
The evaluation includes multiple datasets and considers several aspects such as retrieval quality and robustness.
The use of generated rationales as intermediate signals for document selection and verification is well aligned with recent advances in reasoning-enabled LLMs.
The motivation for replacing fixed top-k retrieval with an adaptive mechanism is clearly articulated.
The paper provides a structured description of the pipeline, including rationale generation, document scoring, and filtering.
Improving the robustness and interpretability of RAG systems is an important problem, particularly for applications in high-stakes domains.
A retrieval pipeline that provides explicit reasoning signals could make RAG systems more transparent and easier to analyze.
The integration of rationale generation into the retrieval selection process represents an interesting combination of reasoning and retrieval techniques.

Weaknesses

The approach heavily depends on the quality of the rationales generated by the language model. Incorrect or biased rationales may negatively affect the retrieval process.
The paper provides limited analysis of how errors in rationale generation propagate through the pipeline.
Some components, such as the adaptive thresholding mechanism, could be described in greater technical detail.
Additional ablation studies would help clarify the relative contribution of rationale generation versus other components.
Although the improvements are promising, the evaluation primarily focuses on benchmark datasets rather than real-world retrieval scenarios.
Rationale-based reasoning and explanation-guided retrieval have been explored in prior work. The paper could better differentiate its approach from related methods in explanation-based retrieval.

---

> ### Author Rebuttal · Authors · 2026-03-31
>
> We thank the reviewer for the constructive feedback and positive assessment of the paper’s motivation, technical plausibility, and overall pipeline design. Several of these points are already discussed in the manuscript, though we agree they can be made easier to see. We clarify them below.
>
> **Rationale quality dependence / error propagation**
>
> Rationale quality is optimized, not assumed. The DPO training described in Section 2.1 treats evidence-selection accuracy as the preference signal: rationales that lead to correct selection are preferred, and those that do not are penalized. This means the model is trained to produce rationales that work for selection, not rationales that merely sound reasonable. When we remove this training (METEORA w/o DPO in Table 2), average recall drops but still remains above the strongest baseline. The degradation is measurable but limited. We will expand the discussion of rationale-error propagation in the revision to make this point more visible.
>
> **Ablations / component isolation**
>
> We respectfully disagree that the paper lacks sufficient ablation evidence, although we agree that it can be presented more clearly. The manuscript already isolates the main components through ablations on rationale refinement, verifier removal, and context expansion, and the appendix further includes chunk-size sensitivity and ECSE interaction analysis. To improve clarity, we will add a unified ablation summary table in the main text. For the detailed component-isolation argument, please also see our response to *Reviewer c72f, “Core contribution / component isolation.”*
>
> **Adaptive thresholding detail**
>
> The full procedure is specified in Section 2.2 and Appendix A.5. Briefly, ECSE computes cosine similarity between each evidence chunk and the pooled rationale embedding, sorts the scores in descending order, and measures the decay using first-order differences. These differences are z-score normalized, and the cutoff is placed at the first statistically significant drop. When no clear drop exists, the method falls back to second-order differences and selects the point of maximum curvature. This is a per-query statistical test on the similarity distribution, not a hand-tuned heuristic. We agree the procedure deserves more prominence in the main text and will add a concise algorithmic box to make it immediately accessible.
>
> **Benchmark datasets vs. real-world relevance**
>
> We would like to clarify that the evaluation is not based on synthetic toy benchmarks. The six datasets are built from real legal, financial, privacy, and academic documents, including contracts, merger agreements, SEC filings, privacy policies, and research papers, and were selected specifically to stress-test long-context evidence selection across domains. We agree that broader deployment evaluation remains important, and we will make that scope clearer in the revision. The current submission also includes a human evaluation aimed at practical auditability of evidence selection, which we view as an important step toward real-world relevance in sensitive settings.
>
> **Differentiation from prior work**
>
> We agree that this distinction can be made sharper in the main paper. The key difference is that prior work typically uses rationales, adaptivity, or verification-related signals in isolation. In contrast, METEORA uses the *same rationales* to support three coupled functions: adaptive evidence selection through ECSE, pre-generation verification through the Verifier, and a human-auditable decision trace. In addition, unlike rationale-guided methods that still retain downstream reranking, METEORA removes the reranking stage and instead performs rationale-driven adaptive selection. We will revise the related work section to make this structural distinction more explicit. For the sharper literature positioning, please also see our response to *Reviewer c72f, “Novelty / distinction from prior work.”*

---

> > ### Author Rebuttal · Reviewer_fqUQ · 2026-04-03
> >
> > Thank you for the detailed response to the rebuttal. I have already given a positive recommendation for the initial score, thus I maintain my original evaluation.

---

### Official Review · Reviewer_c72f · 2026-03-13

**Soundness:** 2
**Presentation:** 3
**Significance:** 2
**Originality:** 2
**Overall Recommendation:** 4
**Confidence:** 4

**Summary:**

This paper proposes a rationale-driven evidence selection framework for RAG in sensitive domains, aiming to replace conventional re-ranking with explicit rationales, adaptive evidence selection, and a verifier module to improve interpretability, efficiency, and robustness to poisoning. The problem setting is important, particularly for high-stakes applications such as legal and financial QA, where transparency and robustness matter. However, while the overall pipeline is reasonably well assembled, I do not find that the current empirical evidence is strong enough to support the paper’s central claims. In particular, the source of the gains is not cleanly isolated, the interpretability claim is not rigorously validated, and the adversarial evaluation is too limited to justify the broader security conclusions. Overall, I think the paper points to a promising direction, but in its current form it is not yet sufficiently validated to be a reliable contribution that others can confidently build on.

**Compliance With Llm Reviewing Policy:**

Affirmed.

**Final Justification:**

The rebuttal addresses my concerns, and other reviewers' opinions also changed my mind.

**Key Questions For Authors:**

1. Can the authors more cleanly isolate the contribution of each component? Right now it is difficult to tell whether the gains come primarily from the rationales, the adaptive cutoff, the verifier, or simply the increased complexity of the full pipeline.
2. How do the authors justify the claim that the generated rationales are faithful explanations rather than post-hoc plausible justifications? Have they considered stronger faithfulness-oriented evaluation protocols?
3. Since efficiency is a central claim, can the authors report end-to-end latency, token consumption, throughput, and deployment cost, rather than using the number of selected chunks as an indirect proxy?
4. Can the authors better clarify the distinction between this work and prior explanation-guided retrieval, self-verification, and multi-stage filtering approaches?

**Limitations:**

The paper discusses some limitations, but the treatment is not yet sufficient. I would encourage the authors to more explicitly discuss that:
1. generated rationales may not be faithful;
2. the verifier itself may be attackable or misled;
3. synthetic poisoning does not fully capture realistic adversarial behavior;

**Strengths And Weaknesses:**

### Strengths
1. The paper addresses an important and practical problem. For high-stakes RAG settings, evidence transparency and robustness are indeed critical concerns, and the motivation is well justified.
2. The proposed method is fairly comprehensive at the system level, combining rationale generation, evidence selection, and verification into a coherent pipeline.
3. The paper attempts to jointly address interpretability, efficiency, and robustness rather than treating them in isolation, which is a worthwhile perspective.

### Weaknesses
1. **The core contribution is not clearly isolated.**
   The method combines several moving parts at once: a DPO-trained rationale generator, rationale-chunk pairing, adaptive thresholding, context expansion, and a verifier. Although the paper includes some ablations, they do not sufficiently disentangle which component is actually responsible for the gains. As a result, the empirical study reads more like a comparison between a more complex pipeline and several baselines, rather than a precise validation of the paper’s main claimed idea.

2. **The “ranking-free” and “interpretable” claims are both overstated.**
   Conceptually, the method still performs evidence prioritization; it simply replaces explicit re-ranking with a more elaborate rationale-based filtering mechanism. Calling this “ranking-free” feels more rhetorical than substantive. Likewise, the paper appears to equate “the model produces rationales” with “the system is interpretable,” which is not sufficient. Generated rationales may be plausible without being faithful to the actual decision process. The current human evaluation does not convincingly establish explanation faithfulness; at best, it suggests that humans can use the provided cues to make local judgments.

3. **The experimental evidence is not strong enough for the breadth of the claims.**
   The paper argues for improvements in precision, robustness, and efficiency, but these dimensions are not all supported equally well. The efficiency claim is especially weak: selecting fewer chunks is only an indirect proxy, and the paper does not provide end-to-end latency, token usage, throughput, or actual system cost. Since the pipeline introduces additional reasoning and verification stages, it is not at all obvious that the overall method is more efficient in practice.

4. **The adversarial robustness evaluation is limited.**
   The poisoning experiments are based on a relatively narrow synthetic attack setup. This is useful as a first test, but far from sufficient to support stronger robustness or security claims. In particular, the paper does not examine adaptive attacks that explicitly target the rationale generator or verifier, nor does it test more realistic retrieval-time or embedding-space poisoning scenarios. At present, the results only show robustness under the authors’ chosen attack protocol, not robustness in a broader sense.

5. **The novelty relative to related work is not fully convincing.**
   The paper positions itself against re-ranking methods, but from a reader’s perspective it still performs evidence prioritization through an intermediate reasoning representation. The distinction from prior work on explanation-guided retrieval, multi-stage filtering, self-verification, or reasoning-augmented selection is not made sharply enough. The overall contribution feels more like a combination of existing ideas than a clearly isolated new methodological advance.

---

> ### Author Rebuttal · Authors · 2026-03-31
>
> We thank the reviewer for constructive feedback. Since several weaknesses and questions overlap, we address the concerns by theme below.
>
> **Core contribution / component isolation**
>
> > **W1 / Q1:** The core contribution is not clearly isolated. Can the authors more cleanly isolate the contribution of each component?
>
> **A1:** METEORA has three main modules: the preference-tuned rationale generator, ECSE, and the verifier. Table 2 already isolates rationale tuning (**METEORA w/o DPO**), verification (**METEORA w/o Verifier**), and context expansion (**METEORA w/ Expansion**). These variants shift metrics differently, showing distinct roles rather than gains from generic pipeline complexity. The adaptive cutoff is supported by ECSE’s statistical thresholding in Section 2.2 and the efficiency analysis in Table 3, where baselines require substantially more evidence to reach comparable recall. Table 4 further isolates the verifier in the poisoning setting. We will revise the text to make this decomposition clearer.
>
> **Interpretability vs faithfulness**
>
> > **W2 / Q2:** The “interpretable” claim is overstated. How do the authors justify the claim that the generated rationales are faithful explanations rather than post-hoc plausible justifications?
>
> **A2:** Our claim is about the interpretability of **evidence selection**, not formal global faithfulness of the model’s full internal reasoning. The intended target is **decision-level / evidence-level interpretability**: whether a human can understand why a piece of evidence was selected, filtered, or flagged. That local decision understanding is exactly what our human evaluation measures. The rationales are also not post-hoc text appended after retrieval: in Section 2.1, rationale quality is tied to whether it leads to correct evidence selection, and at inference the same rationales are used directly by ECSE and the verifier. This is also consistent with the Ethical Considerations paragraph, which notes that users should maintain human oversight and verify that the explanations align with actual behavior.
>
> **Efficiency / practical latency evidence**
>
> > **W3 / Q3:** Since efficiency is a central claim, can the authors report end-to-end latency, token consumption, throughput, and deployment cost?
>
> **A3:** We measured mean *Time-to-First-Token (TTFT)* and *Generation Time* for *SBERT*, *RankRAG*, *METEORA w/o Verifier*, and full *METEORA* on 100 instances from each dataset using the same *NVIDIA L40S GPU (48GB VRAM)*, batch size, and *Llama-3.1-8B-Instruct* backbone. Total time is **4.04s**, **4.61s**, **1.77s**, and **2.91s**, with input tokens **36.89k**, **39.82k**, **18.68k**, and **12.23k**, respectively. TTFT is the dominant cost, consistent with Section 2, and generation time is also lower for METEORA because it conditions on fewer evidence chunks. We did not separately instrument FLOPs; under identical hardware and batch settings, total latency and token usage are the most practical deployment metrics here. Lower total latency therefore also implies lower per-query cost and higher throughput in this controlled setup.
>
> **Robustness scope / attack coverage**
>
> > **W4:** The adversarial robustness evaluation is limited.
>
> **A:** Our robustness claim is scoped to the corpus-poisoning threat model studied in the paper. Following Nazary et al., we poison 30% of QA instances by inserting malicious text into documents that also contain correct context, so poisoned and legitimate evidence co-occur in the same document and cannot be separated by location or semantic similarity alone. Within this setting, METEORA shows consistent gains over both No Defense and Perplexity-based Defense across all six datasets, and the verifier analysis shows that most detected attacks are instruction violations. We agree that adaptive attacks targeting the rationale generator and embedding-space poisoning are important threat vectors not covered here, and we will sharpen the framing in revision to clearly scope the robustness claim to content-level corpus poisoning.
>
> **Novelty / distinction from prior work**
>
> > **W5 / Q4:** Can the authors better clarify the distinction from prior explanation-guided retrieval, self-verification, and multi-stage filtering approaches?
>
> **A4:** We agree that this distinction can be stated more sharply, and we will revise the related work accordingly. Prior methods address only part of the problem: **RAG² / RADIO** use rationales but still rely on downstream reranking; **RankRAG** and **Set-R** still require an LLM to process a retrieved candidate pool at ranking/selection time; **Self-RAG** improves retrieval adaptivity at generation time but does not make evidence selection itself interpretable or perform pre-generation verification. METEORA’s novelty is that it replaces reranking with a unified rationale interface, where the same rationales drive adaptive selection, enable verification, and provide a human-auditable decision trace.

---

> > ### Author Rebuttal · Reviewer_c72f · 2026-04-03
> >
> > I will update my score to 4.

---

### Decision · Program_Chairs · 2026-04-30

**Decision:**

Accept (regular)

**Comment:**

This is the meta-review that summarizes the reviews, rebuttals, and discussion. This paper aims to replace conventional re-ranking with a rationale-guided selection process in RAG. The reviewers agree that the problem is interesting and important when applying RAG in sensitive domains. The proposed METEORA framework and the training process are novel. The authors addressed most of the initially raised concerns during the rebuttal. Please revise the paper according to the review comments in the final version.